# Quantitative Proteomics Analysis of ABA- and GA_3_-Treated Malbec Berries Reveals Insights into H_2_O_2_ Scavenging and Anthocyanin Dynamics

**DOI:** 10.3390/plants13172366

**Published:** 2024-08-25

**Authors:** Germán Murcia, Rodrigo Alonso, Federico Berli, Leonardo Arias, Luciana Bianchimano, Mariela Pontin, Ariel Fontana, Jorge José Casal, Patricia Piccoli

**Affiliations:** 1Fundación Instituto Leloir, Instituto de Investigaciones Bioquímicas de Buenos Aires, CONICET, Buenos Aires C1405, Argentina; lbianchimano@leloir.org.ar (L.B.); jcasal@leloir.org.ar (J.J.C.); 2Instituto de Biología Agrícola de Mendoza, CONICET-Universidad Nacional de Cuyo, Mendoza M5507, Argentina; ralonso@fca.uncu.edu.ar (R.A.); fberli@fca.uncu.edu.ar (F.B.); laarias@mdp.edu.ar (L.A.); afontana@mendoza-conicet.gob.ar (A.F.); ppiccoli@fca.uncu.edu.ar (P.P.); 3INTA-EEA La Consulta, Mendoza M5567, Argentina; pontin.mariela@inta.gob.ar; 4Facultad de Agronomía, CONICET, Instituto de Investigaciones Fisiológicas y Ecológicas Vinculadas a la Agricultura (IFEVA), Universidad de Buenos Aires, Buenos Aires C1053, Argentina

**Keywords:** abscisic acid, gibberellic acid, grapevine, hydrogen peroxide, berry ripening, ROS

## Abstract

Abscisic acid (ABA) and gibberellic acid (GA_3_) are regulators of fruit color and sugar levels, and the application of these hormones is a common practice in commercial vineyards dedicated to the production of table grapes. However, the effects of exogenous ABA and GA_3_ on wine cultivars remain unclear. We investigated the impact of ABA and GA_3_ application on Malbec grapevine berries across three developmental stages. We found similar patterns of berry total anthocyanin accumulation induced by both treatments, closely associated with berry H_2_O_2_ levels. Quantitative proteomics from berry skins revealed that ABA and GA_3_ positively modulated antioxidant defense proteins, mitigating H_2_O_2_. Consequently, proteins involved in phenylpropanoid biosynthesis were downregulated, leading to decreased anthocyanin content at the almost ripe stage, particularly petunidin-3-G and peonidin-3-G. Additionally, we noted increased levels of the non-anthocyanins E-viniferin and quercetin in the treated berries, which may enhance H_2_O_2_ scavenging at the almost ripe stage. Using a linear mixed-effects model, we found statistical significance for fixed effects including the berry H_2_O_2_ and sugar contents, demonstrating their roles in anthocyanin accumulation. In conclusion, our findings suggest a common molecular mechanism by which ABA and GA_3_ influence berry H_2_O_2_ content, ultimately impacting anthocyanin dynamics during ripening.

## 1. Introduction

Grapevines stand as the most economically significant fruit crop worldwide. In Argentina, 92% of the grapevine cultivation area is dedicated to the wine industry, with Malbec being the predominant variety (www.inv.gob.ar). The development and ripening of grape berries involve complex physiological processes marked by dynamic changes in biochemical composition and color. Berry development follows a double sigmoid growth curve with three distinct phases, comprising two periods of growth separated by a lag phase during which cell expansion slows and seeds mature [1]. At the end of the lag phase, a brief period known as veraison indicates the onset of ripening, which is characterized by the rapid accumulation of sugar and anthocyanins in red grape varieties [1]. Polyphenols, particularly anthocyanins, in grape berry skins play a pivotal role in determining red wine quality. In grapevines, polyphenols are classified into two primary groups: non-flavonoids (hydroxybenzoic and hydroxycinnamic acids and their derivatives, stilbenes, and phenolic alcohols) and flavonoids (anthocyanins, flavanols and flavonols) [2]. Generally, they act as phytoalexins, photoprotectants, and potent antioxidants, helping plants to mitigate biotic and abiotic stresses [3], and in grape berry skins, they play a pivotal role in determining sensory attributes and wine quality.

The hormones abscisic acid (ABA) and gibberellins (GAs) are key regulators of berry development and ripening [4]. In grape berries, classified as non-climacteric fruits, the concentration of ABA increases at veraison, influencing the timing of ripening [5], and then declines to low levels. The concomitant increase in ABA levels and berry sugars positively modulates the expression of genes involved in the phenylpropanoid pathway, stimulating the downstream biosynthesis and accumulation of anthocyanins [6,7]. The application of ABA enhances sugar transport to the berries by extending phloem area and up-regulating sugar transport genes, thereby accelerating berry ripening and boosting anthocyanin levels in both wine and table grapes [5,8,9,10,11,12]. 

GAs, along with auxins and cytokinins, promote cell division and expansion during the initial stages of berry development. GA levels in berry tissues are increased during the early stages and then decrease at the initiation of ripening. GAs are primarily synthesized by the seeds, and the final size of the berry depends on the number of seeds [1]. Consequently, the application of gibberellic acid (GA_3_) is commonly employed in seedless grapevine cultivars [13]. Moreover, GAs enhance the sink strength of seeded berries, playing a pivotal role in sugar accumulation [8,14]. However, whilst in the cultivar Malbec, the application of GA_3_ significantly delays the onset of berry ripening and reduces anthocyanin accumulation at veraison [8], in table grapes, it increases polyphenol content in the berry skin [13]. 

Although ABA and GA_3_ are widely used in table grapes to enhance berry color development and sugar accumulation, respectively, their commercial use in wine grapes remains limited due to the lack of a comprehensive understanding concerning their effects [13,15,16,17]. 

Fruit ripening is widely recognized as an oxidative process that requires the turnover of reactive oxygen species (ROS), including free radicals such as hydroxyl radicals (·OH) and superoxide anions (O_2_·^−^), and molecules such as hydrogen peroxide (H_2_O_2_) and singlet oxygen (^1^O_2_) [18,19,20]. In plants, ROS are generated during basal metabolism across various organelles, including mitochondria (aerobic respiration), chloroplasts (photosynthesis), and peroxisomes (photorespiration) [21]. Additionally, several cell-wall- and plasma-membrane-localized enzymes contribute to ROS production, such as NADPH oxidases, amine oxidases, quinone reductases, lipoxygenases, class III peroxidases, and oxalate oxidases [22]. ROS are neutralized by plants’ antioxidative defense mechanisms, which include both enzymatic and non-enzymatic systems. Enzymatic defense involves superoxide dismutase (SOD), catalase (CAT), ascorbate peroxidase (APX), and glutathione peroxidase (GPX), while non-enzymatic defense relies on antioxidants like proline, glutathione, ascorbic acid, carotenoids, and flavonoids [23,24]. Elevated ROS levels exceeding scavenging capacities induce oxidative stress, causing cellular damage and potential death. Conversely, at low levels, ROS function as second messengers in growth, development, and stress responses [25]. The onset of fruit ripening in both climacteric and non-climacteric fruits (grape berries) correlates with H_2_O_2_ accumulation and the modulation of ROS scavenging enzymes, suggesting ROS involvement in fruit development [26,27,28,29,30]. The application of H_2_O_2_ to Kyoho grape berries hastened the accumulation of anthocyanins and total soluble solids (TSS), followed by the up-regulation of genes associated with oxidative stress, cell wall deacetylation, and cell wall degradation [31,32]. There is also evidence of an interplay between ROS and phytohormones like ABA, promoting berry ripening [33], even at the molecular level [34]. However, the link between ABA/GA_3_ and ROS in the regulation of fruit ripening remains largely unexplored.

In this study, we applied exogenous ABA and GA_3_ to the aerial parts of Malbec grapevines to evaluate their effects on berry ripening and anthocyanin dynamics. Physiological, biochemical, and proteomics approaches were used in this study to demonstrate the hypothesis that ABA and GA_3_ modulate H_2_O_2_ levels in berries, consequently influencing berry ripening (anthocyanin and TSS accumulation). We found significant differences regarding berry anthocyanin dynamics among ABA and GA_3_ treatments due to differences in TSS and H_2_O_2_ contents. Moreover, ABA and GA_3_ positively modulated antioxidant defense proteins, reducing berry H_2_O_2_ levels at the almost ripe developmental stage. 

## 2. Results

### 2.1. GA_3_ Promotes BFW and TSS Accumulation during Berry Ripening, Whilst ABA Has No Effect

The impact of ABA and GA_3_ application on Malbec grapevine berries across three developmental stages was investigated (Figure 1A). The GA_3_ treatment increased the overall BFW (berry fresh weight) during the different berry ripening stages, with no significant differences observed between the control and ABA treatments (Figure 1B). ABA-treated berries showed the lowest BFW log_2_ fold change (Log_2_FC) from OOR (onset of ripening developmental stage) to AR (almost ripe developmental stage), and GA_3_-treated berries displayed the lowest BFW fold change from AR to FR (full ripening developmental stage, Figure 1C). The TSS on a per berry basis increased continuously until FR (Figure 1D), and GA_3_ treatment promoted it regardless of the berry ripening stages, while no differences were observed between the control and ABA treatments (Figure 1D). When Log_2_FC TSS was compared among developmental stages, it was observed that ABA- and GA_3_-treated berries displayed significantly reduced TSS accumulation from OOR to AR (Figure 1E). Finally, ABA and GA_3_ treatments showed no impacts on TSS accumulation from AR to FR compared to the control (Figure 1E). 

### 2.2. ABA and GA_3_ Modify the Phenolic Compounds’ Dynamics during Berry Ripening

Whole berry total polyphenols and anthocyanins increased from OOR to FR (Figure 1F,H). ABA-treated berries accumulated the fewest total polyphenols from OOR to AR, and displayed significantly increased accumulation from AR to FR (Figure 1G). In addition, control berries showed the lowest accumulation of total polyphenols from AR to FR (Figure 1G). In relation to total anthocyanins, ABA application likely anticipated the onset of ripening due to the higher content of anthocyanins at OOR. Moreover, ABA treatment showed statistically fewer berry total anthocyanins at AR and a tendency to increase the berry total anthocyanins at FR compared to the control (ABA*DS significant interaction, Figure 1H). In line with this, ABA application decreased the berry total anthocyanin accumulation rate (Log_2_FC) from OOR to AR and increased it from AR to FR (Figure 1I). A similar pattern was observed with GA_3_ treatment, where the accumulation of berry total anthocyanins was significantly less from OOR to AR and higher from AR to FR, compared to the control (Figure 1I). In this case, GA_3_ decreased the berry total anthocyanin content (significant GA_3_*DS interaction) at AR compared to the control (Figure 1H).

### 2.3. ABA and GA_3_ Promote a Shift in Polyphenol Metabolism at AR

Figure 2A shows all of the polyphenols detected in whole berries at AR. ABA treatment decreased the content of the anthocyanins petunidin-3-G, peonidin-3-G, and delphinidin-3-acet and increased the levels of the non-anthocyanins E-viniferin, ferulic acid, myricetin, OH-tyrosol, astilbin, and quercetin. Moreover, ABA decreased the berry total anthocyanin content, while showing no effects on berry total non-anthocyanin and polyphenol content, compared to the control (Figure 2B–D). GA_3_ decreased the contents of the anthocyanins petunidin-3-G, and peonidin-3-G and the non-anthocyanins OH-tyrosol, astilbin, ferulic acid, and myricetin, and only increased the non-anthocyanins E-viniferin and quercetin, compared to the control (Figure 2A). In addition, GA_3_ treatment had no effect on berry total anthocyanins and polyphenols (Figure 2A,C, respectively), whilst it increased the content of berry total non-anthocyanins compared to the control (Figure 2B). Interestingly, both ABA and GA_3_ treatments changed the polyphenol metabolism, promoting the accumulation of non-anthocyanins to the detriment of anthocyanins at AR (Figure 2E). 

### 2.4. ABA and GA_3_ Modify Berry H_2_O_2_ Content Dynamics during Berry Ripening

Figure 3A shows that ABA-treated berries had high berry H_2_O_2_ contents at OOR and markedly reduced berry H_2_O_2_ levels at AR (indicating a significant ABA*DS interaction). This pattern was similar for GA_3_-treated berries. In this sense, GA_3_ application significantly increased the berry H_2_O_2_ content at OOR and then significantly decreased it at AR (GA_3_*DS significant interaction). No statistical differences were found in berry H_2_O_2_ content at FR, either with ABA or GA_3_, compared to the control. In addition, unlike the control, a reduction in berry H_2_O_2_ levels mediated by either ABA or GA_3_ from OOR to AR was observed (Figure 3B). On the contrary, these hormones induced the accumulation of berry H_2_O_2_ from AR to FR compared to the control (Figure 3B).

### 2.5. ABA and GA_3_ Positively Modulate the Enzymatic Antioxidant Defense System 

#### 2.5.1. ABA and GA_3_ Differentially Modulate the Berry Skin Proteome

A total of 1638 proteins were identified and quantified at AR from berry skins, of which 685 were differentially abundant proteins (DAPs) using a cutoff *q*-value of ˂0.05 (Appendix A). Principal component analysis (PCA) of the 1638 proteins confirmed the uniformity of the biological replicates as the three groups of replicates clustered tightly (Figure 4A). The greatest variance in protein abundance was found between control and GA_3_-and ABA-treated samples, as they were separated along the first component (57.3% of the total variance). In particular, the largest difference was evident between the control and the ABA-treated berries. On the other hand, the second principal component (18.3% of the total variance) efficiently separated the ABA-treated berries from the GA_3_-treated ones. The 685 DAPs were used as input queries to perform a k-means clustering heatmap analysis. The most representative GO terms were assigned to each cluster to better visualize the main function of each treatment (Figure 4B). Interestingly, it was found that cluster 1 (96 up-regulated proteins co-expressed by ABA and GA_3_) grouped most of the proteins into the response to H_2_O_2_ (GO: 0042542) and protein glutathionylation (GO: 0010731) categories (Figure 4B and Appendix A). Both GOs are related to mechanisms of oxidative stress alleviation. On the other hand, it was observed that cluster 4 (186 proteins up-regulated only in the control berries) grouped the proteins into the translation (GO: 0006412) and flavonoid biosynthetic pathways (GO: 0009813; Figure 4B and Appendix A). Different functional categories were specifically modulated by ABA and GA_3_ treatments. ABA treatment primarily up-regulated proteins associated with the proteasomal protein catabolic process (GO: 0010498) and photosynthesis (GO: 0015979) categories (cluster 5, size: 159 proteins, Figure 4B). As expected, this hormone also increased the abundance of proteins related to stress response, as suggested by the categories response to reactive oxygen species (GO: 0000302), response to water deprivation (GO: 0009414), response to heat (GO: 0009408), response to oxidative stress (GO: 0006979), response to temperature stimulus (GO: 0009266), response to toxic substance (GO: 0009636), and detoxification (GO: 0098754) (Appendix A). In the case of GA_3_-treated berries, an overrepresentation of the categories tricarboxylic acid metabolism (GO: 0072350) and oxidative photosynthetic carbon pathway (GO: 0009854) was observed (Figure 4B). In addition, this hormone up-regulated the proteins related to aromatic and L-serine and L-glycine amino acid metabolism, as suggested by the categories L-phenylalanine biosynthetic process (GO: 0009094), aromatic amino acid family biosynthetic process (GO: 0009073), glycine biosynthetic process from serine (GO: 0019264), glycine metabolic process (GO: 0006544), L-serine catabolic process (GO: 0006565), serine family amino acid catabolic process (GO: 0009071), L-serine metabolic process (GO: 0006563), serine family amino acid metabolic process (GO: 0009069), and serine family amino acid biosynthetic process (GO: 0009070) (Appendix A). 

#### 2.5.2. ABA and GA_3_ Up-Regulate the Proteins with Antioxidant Functions

ABA and GA_3_ up-regulated the proteins that alleviate oxidative stress and down-regulated those related to flavonoid biosynthetic pathway compared to the control (Figure 4B). Figure 4C shows a PCA biplot mixing the data of DAPs corresponding to antioxidant and anthocyanin biosynthetic pathway enzymes, with the berry total anthocyanin and H_2_O_2_ levels at AR. PC1 explained 74.8% of the variability, while PC2 explained 10.4% of the variability. Regarding antioxidant enzymes, ten differentially abundant enzymes were identified: one peroxidase domain-containing protein (POX), two L-ascorbate peroxidases (APX-1, membrane localized; APX-2, plastid localized), three isoforms of glutathione peroxidase (GPX-1, GPX-2 and GPX-3, cytosol localized), two superoxide dismutases (SOD Mn, mitochondria localized; SOD Cu-Zn, cytosol localized), and two peroxiredoxins (glutaredoxin-dependent peroxiredoxin, Gluta-PRX, and thioredoxine-dependent peroxiredoxin, Thio-PRX) (Appendix A). In relation to anthocyanin biosynthetic pathway enzymes, we found twelve differentially abundant enzymes, including two isoforms of phenylalanine ammonia lyase (PAL-1 and PAL-2), trans-cinnamate 4-monooxygenase (C4H), 4-coumarate-CoA ligase (4CL), 4-coumarate-CoA ligase-Like (4CL-Like), chalcone-flavonone isomerase (CHI), flavonoid 3′-monooxygenase (F3′H), flavonoid 3′,5′-hydroxylase-2 (F3′5′H-2), anthocyanidin synthase (ANS o LDOX), UDP-glucose flavonoid 3-O-glucosyltransferase (UF3GT), UDP-glucose anthocyanidin 5,3-O-glucosyltransferase (UF3,5GT), and anthocyanin acyltransferase (ANAT) (Appendix A and Appendix A). 

Figure 4C shows that all of the antioxidant enzymes except for APX-2 were associated with ABA and GA_3_ treatments, and all enzymes belonging to the anthocyanin biosynthetic pathway were associated with the control. The enzymes GPX-1 and APX-1 were more closely associated with GA_3_-treated berries; meanwhile, the remaining antioxidant enzymes were more closely associated with ABA-treated berries. Moreover, in control berries, a clear association with total anthocyanins and H_2_O_2_ levels was observed.

#### 2.5.3. ABA and GA_3_ Up-Regulate the Oxidative Stress Response Proteins

A more detailed analysis of oxidative stress-related proteins was performed in the 685 DAPs (Appendix A). We found thirteen small heat shock proteins (sHSP-1 to 12 and 22 KDa class IV HSP), five thioredoxins (thioredoxin-1 to -5), one glutaredoxin, and ten glutathion S-transferases (GST-1 to -10). We also found enzymes related to ROS production and membrane degradation caused by oxidative stress: one NADH–ubiquinone reductase, two lipoxygenases (LOX-1 and LOX-2), two phospholipases D (PLD-1 and PLD-2), and one phospholipase C (PLC). Comparing their abundances among treatments by a heatmap, we observed two main clusters (Figure 4D). Cluster 1 grouped all the proteins up-regulated by ABA and GA_3_ (Figure 4D). In this cluster, we found almost (except for five GSTs) all of the proteins that were associated with oxidative stress response and alleviation (sHSP-1 to 12, 22 KDa class IV HSP, thioredoxin-1 to -5 and thioredoxin dependent peroxiredoxin, glutaredoxin, glutaredoxin-dependent peroxiredoxin, GST-3, GST-4, GST-5, GST-6 and GST-8) (Figure 4D). Meanwhile, cluster 2 grouped all the proteins up-regulated almost exclusively by the control (Figure 4D). In this cluster, we found all the enzymes related to ROS production (NADH–ubiquinone reductase, LOX-1, LOX-2), membrane degradation (LOX-1, LOX-2, PLD-1, PLD-2 and PLC), and five GSTs (GST-1, GST-2, GST-7, GST-9 and GST-10). Finally, noting that the enzymes involved in membrane peroxidation were up-regulated in the control berries, we decided to assess the content of MDA, a product of lipid peroxidation. Contrary to expectations, despite the upregulation of enzymes involved in membrane lipid peroxidation, control berries exhibited significantly lower MDA content compared to those treated with hormones (Appendix A). 

### 2.6. Berry Total Anthocyanins Accumulation Depends on Berry Sugar and H_2_O_2_ Content 

A linear mixed-effects model showed that the fixed effects “TSS” and “H_2_O_2_” were statistically significant with high predictive accuracy in the entire model (R^2^ = 0.98 and *p* ˂ 0.0001) (Appendix A and Figure 5). In addition, the total fixed effects explained 95% of the variance, out of a total of 98% explained by the model including both fixed and random effects (Appendix A). In this sense, the random effect “Treatment” only explained 3% of the variance (Appendix A). This result demonstrated that berry total anthocyanin accumulation positively depended on berry sugar and H_2_O_2_ contents. 

## 3. Discussion

Our results show significant variations in H_2_O_2_ levels among the ripening stages and between ABA and GA_3_ treatments, subsequently impacting the dynamics of total anthocyanin accumulation throughout berry development. Importantly, the observed differences among treatments were attributed to the effects of TSS and H_2_O_2_ (R^2^ = 0.98). Thus, berry total anthocyanins positively depended on berry sugar and H_2_O_2_ contents. Both hormone treatments reduced berry H_2_O_2_ content at AR, and this was associated with an increased abundance of proteins related to the antioxidant enzyme system. Finally, this reduction in berry H_2_O_2_ levels was correlated with a downregulation of the abundance of enzymes belonging to the anthocyanin biosynthesis pathway observed in the hormone treatments.

GA_3_ is widely recognized in the table grape industry for its role in enhancing both yield and sugar content, especially in seedless berries [13], but is also effective in seeded wine grapes [8,14]. The present study provides additional evidence of GA_3_’s impact on berry physiology, increasing sugar accumulation on a per berry basis and berry growth during ripening, possibly as an enhancer of sink strength [8]. Interestingly, proteomic analysis unveiled the GA_3_-induced upregulation of two proteins associated with cell wall softening and fruit ripening as pectin esterase (F6HJZ5) and pectin acetylesterase (D7TFE6; [35,36]). However, GA_3_ treatment downregulated two expansins, expansin (E0CQY0) and expansin-B2 (D7SLR0), which are also linked to berry ripening, suggesting a complex mechanism regulating the berry growth by GA_3_. 

The role of ABA in promoting the accumulation of anthocyanins in grape berries is well-established, primarily through the upregulation of the *PHENYLALANINE AMMONIA-LYASE-1*, *PHENYLALANINE AMMONIA-LYASE-2* (*VvPAL-1*, *VvPAL-2*), *CHALCONE SYNTHASE* (*VvCHS*), *FLAVONONE-3-HYDROXYLASE* (*VvF3H*), *FLAVANONE-3-HYDROXYLASE* (*VvFHT*), *GLUCOSE ACYLTRANSFERASE* (*VvAT*), and *GLUTATHIONE S-TRANSFERASE* (*VvGST*) structural genes, as well as the *MYB-RELATED TRANSCRIPTION FACTORS* (*VvMYBA1*, *VvMYBA2*, *VvMYBPA1*) regulatory genes, and the abundance of proteins associated with the phenylpropanoid biosynthesis pathway [37,38,39,40]. In the present study, ABA-treated berries exhibited a significant increase in anthocyanin content at OOR, highlighting the role of ABA as a major regulator of grape berries at early stages of ripening [5,10]. As berry maturity progressed, ABA treatment showed statistically less berry total anthocyanins at AR and a tendency to increase the berry total anthocyanins at FR. This dynamic variation between ABA-treated and control berries paralleled differences in H_2_O_2_ dynamics during berry ripening. Thus, we demonstrated that ABA treatment modulated the synthesis and degradation of H_2_O_2_, thereby influencing the overall accumulation of berry total anthocyanins during ripening. These results are consistent with those indicating that ABA regulates ROS generation and accumulation by modulating the activity of NADPH oxidase, the primary enzyme catalyzing ROS generation in the apoplast, and inducing the degradation of H_2_O_2_ through the upregulation of the *OsCATB* gene which encodes for the antioxidant enzyme catalase B in rice leaves [41,42,43]. 

Similar to the results observed for ABA treatment, GA_3_ application significantly reduced total anthocyanins at AR, followed by an increase in their accumulation at FR stage. Again, these results were correlated with fluctuations in H_2_O_2_ levels during berry ripening, with lower levels of H_2_O_2_ at AR and higher accumulation of it from AR to FR. Previous studies have demonstrated the role of GA_3_ in promoting antioxidant enzyme activities to maintain redox homeostasis under various environmental stresses [44]; however, our study represents the first report on the ability of GA_3_ to induce ROS generation. Unlike ABA treatment, the presence of high levels of H_2_O_2_ at OOR failed to promote a significant increase in total anthocyanin content in GA_3_-treated berries, a result that will be discussed in detail later in this section.

Gambetta et al., 2010 [6] demonstrated that grapevine orthologs of key sugar and ABA-signaling components are intricately regulated by the interplay between sugar and ABA, affecting the accumulation of total anthocyanins. They specifically evaluated Cabernet Sauvignon berries in two experimental systems: field-grown (deficit-irrigated) and cultured with sucrose and ABA. They found that the expression of the *VvMYBA1* gene, a crucial transcription factor responsible for activating anthocyanin biosynthesis, was significantly upregulated by sugars in the presence of ABA. Furthermore, Hung et al., 2008 [45] demonstrated that using a chemical trap for H_2_O_2_, dimethylthiourea, inhibits the ABA-induced accumulation of anthocyanins in rice leaves. Additional evidence suggests that the application of ABA or H_2_O_2_ to grapevine berries can accelerate the onset of ripening by upregulating the expression of phenylpropanoid biosynthesis pathway genes [32,39]. These findings collectively suggest that H_2_O_2_ acts downstream of ABA signaling, with both H_2_O_2_ and sugars acting as major regulators of total anthocyanin synthesis and accumulation. Consistent with these reports, our study revealed a dependence of berry total anthocyanin content during ripening on TSS and H_2_O_2_ levels (R^2^ = 0.98). ABA-treated berries at OOR exhibited higher total anthocyanin levels compared to the control, attributed to elevated H_2_O_2_ content on a per berry basis. Then, ABA application resulted in lower berry total anthocyanin accumulation from OOR to AR, corresponding to reduced TSS and H_2_O_2_ accumulation relative to the control. Lastly, a significant increase in berry total anthocyanin accumulation was observed from AR to FR with ABA applications, which correlated with increased H_2_O_2_ accumulation compared to the control. Meanwhile, GA_3_ treatment, despite inducing the highest levels of H_2_O_2_ and TSS on a per berry basis at OOR, did not result in higher berry total anthocyanin content compared to ABA and control treatments, a pattern also evident at FR. These observations agree with previous studies by Loreti et al., 2008 [46], which demonstrated that sucrose-induced activation of the anthocyanin synthesis pathway was repressed by GA in Arabidopsis seedlings. In addition, recent findings by An et al., 2024 [47] suggest that GA may act as a repressor of anthocyanin synthesis by promoting the transcription and stability of MdbHLH162, which negatively regulates anthocyanin biosynthesis by disrupting the formation of anthocyanin-activated MdMYB1-MdbHLH3/33 complexes in apple. According to this, it is possible that GA_3_ is acting as a repressor of anthocyanins synthesis in Malbec berry skins. 

Observing the reduction in berry H_2_O_2_ and total anthocyanins levels following ABA and GA_3_ treatments at the AR stage, a proteomic analysis was conducted to unravel a shared molecular mechanism governing this effect in grape berries. Both ABA- and GA_3_-upregulated proteins were associated with the antioxidant enzymatic system, possibly alleviating H_2_O_2_ levels. Specifically, increased abundances of ROS-scavenging enzymes, such as POX, SOD, GPX, APX, and peroxiredoxins, in response to both treatments were observed. This aligns with previous studies in which exogenous applications with ABA or GA_3_ across various plant species, including grapevine [48,49], rice [43], and maize [50,51], increased ROS scavenging enzymes activities under abiotic stresses. Furthermore, GA_3_ treatments increased the activities of the antioxidant enzymes POX and SOD in *Phellodendron chinensis* seedlings growing at optimal conditions [52]. However, the modulation of H_2_O_2_ levels by the enzymatic system in grapevine berries or other fruits regarding these hormones remains unexplored in the existing literature. Our analysis further revealed that ABA treatment led to an augmentation in proteins associated with the GO term photosynthesis (GO:0015979), particularly those related to photosystem II and ATP synthesis in the chloroplast, such as oxygen-evolving enhancer protein 1 (F6I229), oxygen-evolving enhancer protein 3-2 (F6H8B4), chlorophyll a-b binding protein (F6HKS7), 23 kDa subunit of oxygen evolving system of photosystem II (A5B1D3), PsbP C-terminal domain-containing protein (E0CQM8), photosystem II stability/assembly factor (D7T9G8), Ferredoxin (F6HK77), and ATP synthase delta chain (F6HVW3). This suggests that the sustained functioning of the thylakoid electron transport chain from H_2_O to NADP^+^ reduces the likelihood of ROS generation. Accordingly, this could be one of the reasons for the differences observed between ABA and GA_3_ regarding H_2_O_2_ content at AR. ABA and GA_3_ treatments induced an increase in proteins abundance linked to oxidative stress responses, including small heat shock proteins (sHSPs), thioredoxins, and glutaredoxin, known for their multifaceted roles in mitigating oxidative stress [53,54,55]. Moreover, both treatments increased the protein abundance of five glutathione S-transferases (GSTs), known for their involvement in detoxification processes and in the attenuation of oxidative stress [56]. However, three GSTs (GST-1, GST-9 and GST-10) exhibited higher protein abundance in the control treatment, possibly indicating their role in anthocyanin transport from the endoplasmic reticulum to the vacuoles, as suggested by Sun et al., 2016 [57], given the higher accumulation of anthocyanins observed in this treatment at AR. Regarding ROS generation, in our proteomic dataset, we could not identify and quantify the major enzyme that catalyzes the production of H_2_O_2_ in the apoplast, NADPH oxidase. Instead, our attention was focused on NADH–ubiquinone reductase, a protein believed to be a significant source of ROS within mitochondria, contributing substantially to cellular oxidative stress [22,58]. Furthermore, our analysis revealed the presence of two lipoxygenases, LOX-1 and LOX-2, implicated in catalyzing membrane lipid peroxidation and subsequent liberation of H_2_O_2_ from the enzyme surface, being potential ROS-generating enzymes [22,59]. Given the upregulation of NADH–ubiquinone reductase, LOX-1, and LOX-2 in control berry skins, we suggested that the increased activity of these enzymes accounted for the elevated H_2_O_2_ levels observed at AR in this treatment. Considering the increased protein abundance of LOX-1, LOX-2, and two phospholipases D (PLD-1 and PLD-2), higher levels of MDA, a well-known indicator of membrane peroxidation, might be expected. However, our hormonal treatments increased MDA content at AR compared to the control, suggesting that H_2_O_2_ may serve primarily as a signaling molecule rather than a toxic byproduct, thereby avoiding damage to cellular membranes, as suggested by Xi et al., 2017 [60]. Another possible explanation for this result is that the ABA and GA_3_ treatments may have increased H_2_O_2_ levels at the OOR stage, potentially through the upregulation of NADH–ubiquinone reductase, LOX-1, and LOX-2. This increase could have led to elevated membrane lipid peroxidation. As a result, MDA levels might have remained high in hormone-treated berries until the AR stage.

In summary, berries treated with ABA and GA_3_ exhibited reduced levels of H_2_O_2_ at AR, attributed to the upregulation of ROS-scavenging proteins and the downregulation of ROS-generating proteins. Both treatments decreased anthocyanin content at AR, particularly petunidin-3-G and peonidin-3-G, correlated with a downregulation of the abundance of enzymes belonging to the anthocyanin biosynthesis pathway. However, both ABA and GA_3_ treatments increased the protein abundance of the transcription factor Abscisic acid stress-ripening protein 2 (ASR2: F6GY46) at AR. ASR mediates glucose–ABA and glucose–GA crosstalk, modulating sugar accumulation and fruit ripening [61,62]. Accordingly, the up-regulation of ASR2 may have enhanced berry total anthocyanins accumulation from AR to FR observed in the ABA- and GA_3_-treated berries. ABA and GA_3_ treatments prompted a metabolic shift from anthocyanin to non-anthocyanin biosynthesis at AR. In this sense, both ABA and GA_3_ treatments led to increased levels of the stilbene E-viniferin and the flavonol quercetin compared to the control. Interestingly, these molecules exhibit potent antioxidant properties, surpassing even those of resveratrol [63]. Furthermore, it has been demonstrated that the specialized structure of quercetin, comprising a free 3-OH group and 3′,4′-catechol, provides it with antioxidant properties, further helping in quenching the ROS generated by cells (reviewed in Singh et al., 2021 [64]). Consequently, the accumulation of E-viniferin and quercetin may boost ROS scavenging, thereby reducing the H_2_O_2_ content at AR observed in ABA- and GA_3_-treated berries.

## 4. Materials and Methods

### 4.1. Plant Material and Experimental Conditions 

The experiment was carried out during the 2016-2017 growing season in a commercial vineyard (La Pirámide, Catena Zapata winery; 33°09′58″ S, 68°54′31″ W and 1000 m asl, Mendoza, Argentina). Grapevines of a selected clone of *Vitis vinifera* L. cv. Malbec planted on their own roots were used. The vines were trained on a vertical trellis system arranged in north-south-oriented rows (2 m row spacing and 1.20 m between plants) and were maintained without soil water restriction using a drip irrigation system. The vines were cane-pruned and shoot-thinned to 12 shoots per vine and two clusters per shoot. The assay was set in a random design with three treatments (control, ABA and GA_3_) and three biological replicates. The biological replicate consisted of 5 plants from 7 consecutive plants in the row. Each biological replicate was sampled at the onset of ripening (OOR, stage 35), almost ripe (AR, stage 37) and full ripening (FR, stage 38) based on Coombe et al. [65] (Figure 1A). ABA, GA_3_, and water (control) solutions were sprayed with a hand-held sprayer onto the aerial parts of the plant (leaves and bunches) until runoff, with a 14-day frequency from the berry pea-size stage (stage 31, Figure 1A) and during late afternoon to minimize ABA photodegradation. Treatment doses were as follows: 1 mM ABA (± -S-*cis*, *trans* abscisic acid, PROTONE SL, Valent BioSciences, Libertyville, IL, USA), 1 mM GA_3_ (GIBERELINA KA, S. Ando & Cía. SA, Buenos Aires, Argentina), and water (control). All solutions were supplemented with 0.05% (*v*/*v*) Triton X-100 as a surfactant. 

### 4.2. Berry Sampling, Fresh Weight and Total Soluble Solids 

At each developmental stage (OOR, AR and FR), two days after the hormone application, 60 berries per biological replicate were randomly collected from 10 clusters (6 berries from each cluster: 2 top, 2 middle, and 2 bottom berries). The berries were placed in nylon bags and kept on dry ice to prevent protein degradation and dehydration. In the laboratory, 32 berries from each biological replicate were separated to measure berry fresh weight (BFW, g berry^−1^) and total soluble solids (TSS, °Brix). To achieve this, 32 berries were put into nylon bags and crushed via hand pressing, and the TSS was measured in the juice with a Pocket PAL-1 digital hand-held refractometer (Atago Co., Ltd., Tokyo, Japan). Then, the °Brix was multiplied by the BFW to express TSS in sugar on a per berry basis (mg berry^−1^). The remaining 28 berries from each biological replicate were stored at −80 °C for further analysis.

### 4.3. Berry Phenolic Extraction, Berry Total Polyphenols and Anthocyanins 

Fifteen berries per biological replicate were deseeded and ground into a fine powder in liquid nitrogen using a mortar and pestle. One gram of the powder was then macerated with 10 mL of 1% HCl-methanol solution. The extraction was performed by heating the samples at 70 °C for 1 h, followed by three rounds of 5 min sonication in darkness. Then, the samples were centrifuged for 10 min at 5000× *g* and the supernatant was collected for spectrophotometric measurements and chromatographic analysis. Absorbance of the extracts was read at 280 or 520 nm for the determination of total polyphenols and total anthocyanins, respectively, according to Berli et al., 2008 [66], with a Cary-50 UV-Vis spectrophotometer (Varian Inc., Palo Alto, CA, USA). Finally, the results were expressed on a per berry basis.

### 4.4. Berry Phenolic Compounds Profile 

Anthocyanins and non-anthocyanins (low molecular weight polyphenols, LMWP) were analyzed using high-performance liquid chromatography with a diode array and fluorescence detection (HPLC-DAD-FLD, Dionex Ultimate 3000 system, DionexSoftron GmbH, Thermo Fisher Scientific Inc., Germering, Germany). Anthocyanin determination was performed at AR according to Urvieta et al., 2018 [67] with minor adjustments. Briefly, a 500 μL aliquot of berry phenolic extract (described for spectrophotometric measurements) was dried via vacuum centrifugation and dissolved in 1 mL of initial mobile phase prior to chromatographic analysis. Anthocyanins were separated in a reversed-phase Kinetex C18 column (3.0 × 100 mm, 2.6 μm) Phenomenex (Torrance, CA, USA). The mobile phase was composed of ultrapure H_2_O/FA (formic acid)/MeCN (acetonitrile) (87:10:3 *v*/*v*/*v*; eluent A) and ultrapure H_2_O/FA/MeCN (40:10:50 *v*/*v*/*v*; eluent B). The separation gradient was as follows: 0 min, 10% B; 0–10 min, 25% B; 10–15 min, 31% B; 15–20 min, 40% B; 20–30 min, 50% B; 30–35 min, 100% B; 35–40 min, 10% B; 40–47 min, 10% B. The mobile phase flow, column temperature, and injection volume were 1 mL min^−1^, 25 °C, and 5 μL, respectively. Quantification was carried out by measuring peak area at 520 nm and the content of each anthocyanin was expressed as malvidin-3-glucoside equivalents, using an external standard calibration curve (1–250 mg L^−1^, R^2^ = 0.997). The identity of detected anthocyanins was confirmed by comparison with the elution profile and identification of analytes performed in previous research [68]. Then, the results were expressed on a per berry basis. For LMWP compounds, berry phenolic extracts were analyzed according to analytical conditions reported by Ferreyra et al., 2021 [69], using the same column as for anthocyanins. The mobile phase was an aqueous solution of 0.1% FA (solvent A) and MeCN (solvent B). The gradient was as follows: 0–1.7 min, 5% B; 1.7–10 min, 30% B; 10–13.5 min, 95% B; 13.5–15 min, 95% B; 15–16 min, 5% B; 16–19, 5% B. The total flow rate was set at 0.8 mL min^−1^. The column temperature was 35 °C and the injection volume was 5 μL. The identification of LMWP was based on the comparison of the retention times of phenolic compounds in samples with those of authentic standards. Standards of (+)-catechin (≥99%), (−)-epicatechin (≥95%), (+)-procyanidin B1 (≥90%), procyanidin B2 (≥90%), (−)-epigallocatechin (≥95%), (−)-gallocatechin gallate (≥98%), (−)-epicatechin gallate (≥95%), polydatin (≥95%), piceatannol (≥95%), (+)-E-viniferin (≥95%), quercetin hydrate (95%), quercetin 3-β-d-galactoside (≥97%), quercetin 3-β-d-glucoside (≥90%), kaempferol-3-glucoside (≥99%), myricetin (≥96%), naringin (≥95%), 3-hydroxytyrosol (≥99.5%), caftaric acid (≥97%), ferulic acid (≥99%), gallic acid (99%), and phlorizin (≥99%) were purchased from Sigma-Aldrich (St. Louis, MO, USA). External calibration was used as a quantification approach. Linear ranges between 0.05 and 40 mg L^−1^ with a coefficient of determination (R^2^) higher than 0.993 were obtained. The results were expressed on a per berry basis. HPLC-grade MeCN and FA were sourced from Mallinckrodt Baker Inc. (Pillispsburg, NJ, USA). Ultrapure water was procured from a Milli-Q system (Millipore, Billerica, MA, USA).

### 4.5. Berry Hydrogen Peroxide Content and Lipid Peroxidation 

Measurements were performed according to Junglee et al., 2014 [70], with some modifications. Briefly, 150 mg of deseeded berry frozen powder was homogenized with 1 mL of trichloroacetic acid (TCA; Sigma-Aldrich Corp., St. Louis, MO, USA) buffer (0.25 mL TCA 0.1% (*w*/*v*), 0.5 mL KI 1 M, and 0.25 mL K phosphate buffer 10 mM pH 8)) at 4 °C for 10 min. The homogenate was centrifuged for 15 min at 12,000× *g* at 4 °C. The supernatant was collected and incubated for 20 min at room temperature. The absorbance of the extracts was read at 350 nm with a Cary-50 UV-Vis spectrophotometer. A calibration curve obtained with H_2_O_2_ standard solutions (100 vol., 30%) prepared in TCA buffer was used for quantification (8–100 nmol of H_2_O_2_, R^2^ = 0.999). Then, the results were expressed on a per berry basis.

Malondialdehyde (MDA) content was measured following the procedure described by Beligni and Lamattina 2002 [71]. For that, 100 mg of deseeded berry frozen powder were suspended in 2 mL of stock solution (15%, *w*/*v* TCA, 0.5%, *w*/*v* thiobarbituric acid (TBA, Sigma-Aldrich Corp., St. Louis, MO, USA), and 0.25%, *w*/*v* hydrochloric acid (37%)). The mixture was stirred vigorously and incubated at 95 °C for 60 min. The samples were centrifuged at 9300× *g* for 10 min, the supernatant was collected, and the absorbance of the extracts was measured at 535 nm. The concentration was calculated considering an MDA molar extinction coefficient = 1.56 × 10^5^ M^−1^ cm^−1^.

### 4.6. Proteomic Analysis Using High-Resolution Mass Spectrometry 

#### 4.6.1. Berry Skin Protein Extraction and Quantification

The protein fraction was extracted from berry skins at AR using the method previously described by Negri et al., 2008 [72] with some modifications. Thirteen berries per biological replicate (*n* = 3) were peeled, and berry skins were finely powdered in liquid nitrogen using a pestle and mortar. Two grams of the powder were then resuspended in 10 mL of extraction buffer (0.7 M sucrose (Sigma-Aldrich Corp., St. Louis, MO, USA), 0.5 M Tris-HCl pH 8 (MP Biomedicals, California, USA), 10 mM disodium EDTA salt (Promega, Madison, WI, USA), 1 mM PMSF (phenylmethylsulfonyl fluoride, Sigma-Aldrich Corp., St. Louis, MO, USA), 0.2% (*v*/*v*) β-mercaptoethanol (Sigma-Aldrich Corp., St. Louis, MO, USA), protease inhibitor cocktail (Sigma-Aldrich Corp., St. Louis, MO, USA), and PVPP (Sigma-Aldrich Corp., St. Louis, MO, USA)) and shaken for 10 min at 4 °C. Proteins were extracted by the addition of an equal volume of ice-cold Tris-buffered phenol pH 8 (Sigma-Aldrich Corp., St. Louis, MO, USA). The sample was shaken for 30 min at 4 °C, incubated for 2 h at 4 °C, and finally centrifuged at 5000× *g* for 20 min at 4 °C to separate the phases. Then, 9 mL of the upper phenol phase was collected, and the proteins were precipitated by the addition of 40 mL of ice-cold 0.1 M ammonium acetate in methanol. The sample was vortexed briefly and maintained at −20 °C overnight. Precipitated proteins were recovered by centrifuging at 13,000× *g* for 30 min at 4 °C, then washed again with cold methanolic ammonium acetate and three additional times with cold 80% (*v*/*v*) acetone. The final pellet was dried at room temperature and resuspended in 500 µL of buffer (7 M urea (Promega, Madison, WI, USA), 2 M thiourea (Sigma-Aldrich Corp., St. Louis, MO, USA), 4% (*v*/*v*) IGEPAL (Sigma-Aldrich Corp., St. Louis, MO, USA), and 50 mg mL^−1^ DTT (Sigma-Aldrich Corp., St. Louis, MO, USA)). Finally, the sample was centrifuged at 13,000× *g* for 3 min and the supernatant was stored at −80 °C until it was used for protein analysis. The protein concentration was determined by the Bradford assay (Bio-Rad Laboratories Inc., Hercules, CA, USA).

#### 4.6.2. Nano-LC-Orbitrap Tandem Mass Spectrometry (Nano-LC-MS/MS) Protein Analysis 

Fifty micrograms of total protein was boiled (95 °C, 5 min) in Laemmli buffer (0.0625 M Tris base (Promega, Madison, WI, USA), 0.07 M Sodium Dodecyl Sulfate (SDS, Promega, Madison, WI, USA), 10% (*v*/*v*) glycerol (Promega, Madison, WI, USA), 5% β-mercaptoethanol (Sigma-Aldrich Corp., St. Louis, MO, USA), and 0.005% bromophenol blue (Sigma-Aldrich Corp., St. Louis, MO, USA)) and run in 10% SDS-PAGE. The proteins were allowed to run only 2 cm into the separating gel. The gel was stained according to the colloidal Coomassie Brilliant Blue G-250 (Sigma-Aldrich Corp., St. Louis, MO, USA) procedure [73], and the dried fragments of the gel corresponding to each biological replicate were sent to the Proteomics Core Facility CEQUIBIEM, Buenos Aires, Argentina. Proteins were reduced with 10 mM DTT for 45 min at 56 °C and alkylated with 50 mM iodoacetamide (Sigma-Aldrich Corp., St. Louis, MO, USA) for 45 min in darkness. Proteins were digested overnight with sequencing-grade modified trypsin (Promega, Madison, WI, USA). Then, the samples were lyophilized and resuspended with 30 µL of 0.1% trifluoroacetic acid (Sigma-Aldrich Corp., St. Louis, MO, USA). Zip-Tip C18 (Merck Millipore, Burlingtone, MA, USA) columns were used for desalting. The resulting peptides were separated in a nano-HPLC (EASY-nLC 1000, Thermo Fisher Scientific, Germering, Germany) coupled with a mass spectrometer with Orbitrap technology (Q-Exactive with High Collision Dissociation cell and Orbitrap analyzer, Thermo Fisher Scientific, Germany). Peptides were ionized by electrospraying (EASY-SPRAY, Thermo Scientific, Germany) at a voltage of 1.5 to 3.5 kV.

### 4.7. Proteomics Data Analysis 

Proteome Discoverer 2.2 software (ThermoScientific, Germany) was used to match the identity of peptides to the grapevine reference proteome set from uniprot (Vitis vinifera-UP000009183-Uniprot). The raw intensity values obtained from the MS data were normalized using the total ion current normalization method. The critical search parameters were as follows: precursor ion mass tolerance of 10 ppm, fragment mass tolerance of 0.05 Da, trypsin enzyme with a maximum of two missed cleavages allowed, variable modifications including oxidation and carbamidomethylation of cysteine, and a minimum of two peptides identified per protein. Missing values in the dataset were calculated via the principal component method using the package *missMDA* in R. Only proteins with high confidence and a percolator *q*-value lower than 0.01 were considered for the one-way ANOVA analysis. 

### 4.8. Statistical and Bioinformatic Analysis 

Data were analyzed using the software platform R 4.1.1. A multi-factor ANOVA was used to evaluate the effects of ABA, GA_3_, developmental stage, and their interactions on BFW, TSS, berry total polyphenols, berry total anthocyanins, and berry H_2_O_2_ content kinetics using the aov function. For the analysis of Log_2_ fold change, berry anthocyanin and non-anthocyanin profiles, and MDA relative amount, one-way ANOVA followed by Fisher’s least significant difference (LSD) test was applied using the aov function. Principal component analysis (PCA) was performed using the function prcomp from the factoextra package in R. The effects of TSS and H_2_O_2_ content on berry total anthocyanin accumulation across the three developmental stages were investigated via a linear mixed-effects model. “Treatments” was considered as a random effect on the intercept and slopes, while “TSS” and “H_2_O_2_” were considered as fixed effects. The linear mixed-effects regression was fitted using the function lmer from the lme4 R package. The *p*-values of the fixed effects were obtained with the function anova from the lmerTest package in R. Total variance explained by the model was partitioned with the function r.squaredGLMM from the MuMIn R package in order to estimate the fraction of variance explained by the fixed and random effects. To identify differentially abundant proteins (DAPs), one-way ANOVA was performed with a significance threshold of *p* < 0.05 and *q* < 0.05 (for multiple comparisons), using the aov function and the qvalue package. The heatmaps used for treatment comparison were designed using the pheatmap package. Bubble plots showing GO terms were designed using the ggplot2 package and custom R scripts. Selected GO terms for biological processes were retrieved from STRING functional enrichment of stringApp 2.0.3 [74] in the Cytoscape 3.10.1 open software platform [75] with the grapevine reference genome as background and avoiding redundancy. The main network of each cluster, from the *k*-means clustering heatmap, was re-clustered with the MCL algorithm in stringApp for Cytoscape, with an inflation value of 4. In this sense, each heatmap cluster was labeled with the most representative GO term for biological function, which was retrieved by setting the redundancy cutoff to 0 in STRING functional enrichment. 

## 5. Conclusions

To our knowledge, this is the first report showing that ABA and GA_3_ modify the H_2_O_2_ levels during grapevine berry ripening. We demonstrated how these hormones modulate the dynamics of berry total anthocyanins by regulating TSS and H_2_O_2_ levels across each stage of berry development (OOR, AR and FR). The diminished H_2_O_2_ levels in ABA- and GA_3_-treated berries at AR were attributed to both elevated levels of ROS-scavenging and oxidative stress response proteins, as well as decreased levels of ROS-generating proteins. Furthermore, the increased accumulation of E-viniferin and quercetin in ABA- and GA_3_-treated berries likely strengthened the H_2_O_2_ scavenging activity. However, further investigations are necessary to uncover the molecular mechanisms underlying ROS generation in ABA and GA_3_-treated berries during ripening. This study provides valuable insights into how ABA and GA_3_ affect grape ripening and quality in a wine cultivar, with potential implications for viticulture and wine production. In this sense, we propose that ABA and GA_3_ applications could serve as effective technological tools for regulating berry ripening and berry anthocyanin accumulation in commercial vineyards of wine cultivars.

## Figures and Tables

**Figure 1 plants-13-02366-f001:**
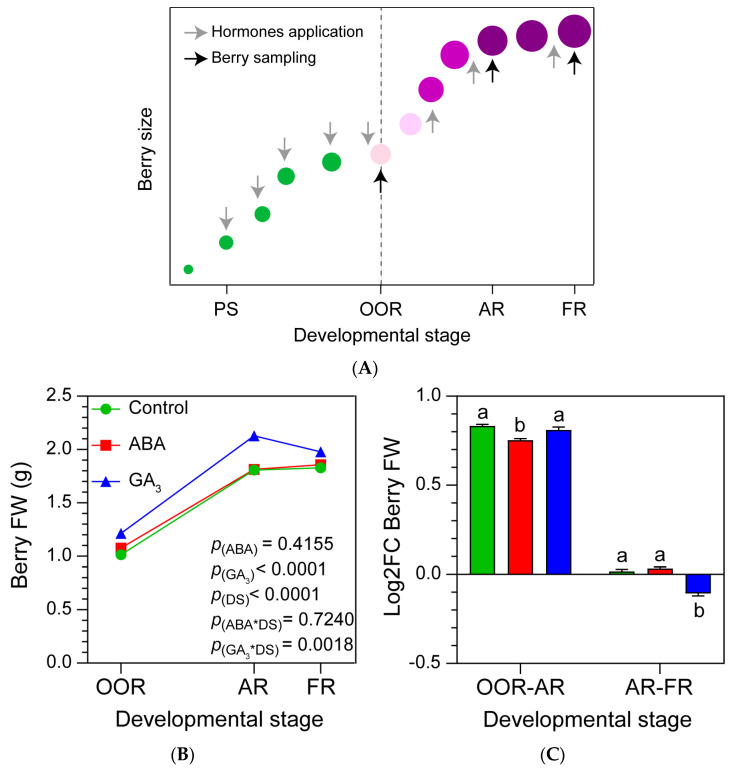
(**A**) Scheme of hormone application and berry sampling during berry growth and development. Hormones were applied every two weeks starting at PS. PS: berry pea-size stage; OOR: onset of ripening stage AR: almost ripe stage; FR: full ripening stage. Vertical dashed line indicates the veraison stage. (**B**) Berry fresh weight; (**D**) total soluble solids per berry; (**F**) total polyphenols per berry; (**H**) total anthocyanins per berry according to the developmental stage and treatment. (**C**,**E**,**G**,**I**) Log_2_ fold change in each variable from OOR to AR (OOR-AR) and from AR to FR (AR-FR). Values are means ± SEs, *n* = 3. Some errors cannot be shown because the SEs are smaller than the symbol. *p*(ABA), *p*(GA_3_), and *p*(DS): effects of ABA, GA_3_, and developmental stage, respectively; *p*(ABA*DS) and *p*(GA_3_*DS): interaction effects of factors. One- and two-way ANOVA followed by Fisher’s LSD test were applied. Different letters indicate significant differences (*p* < 0.05).

**Figure 2 plants-13-02366-f002:**
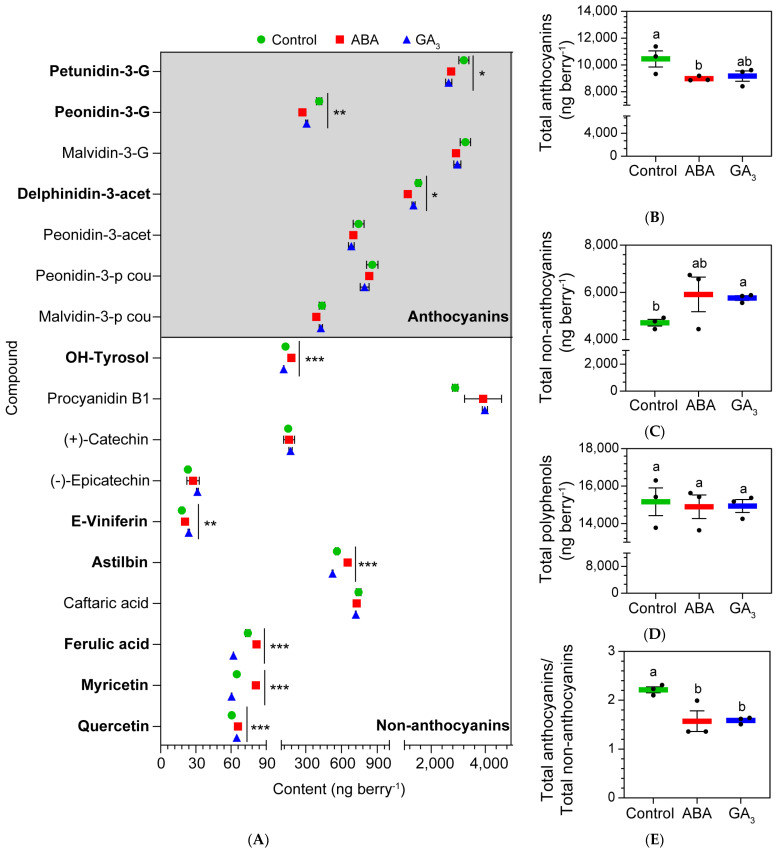
(**A**) Anthocyanins and non-anthocyanins (low mass weight polyphenols) found in berries at the almost ripe stage (AR); (**B**) total anthocyanins per berry at AR; (**C**) total non-anthocyanins per berry at AR; (**D**) total polyphenols per berry at AR; (**E**) total anthocyanins/total non-anthocyanins ratio per berry at AR. Values are means ± SEs, *n* = 3. Some errors cannot be shown because the SEs are smaller than the symbol. One-way ANOVA followed by Fisher’s LSD test was applied. Different letters indicate significant differences (*p* < 0.05). Significance codes: (***) *p* ˂ 0.001; (**) *p* ˂ 0.01; (*) *p* ˂ 0.05. G: glucoside; acet: acetylglucoside; p cou: p-coumaroylglucoside.

**Figure 3 plants-13-02366-f003:**
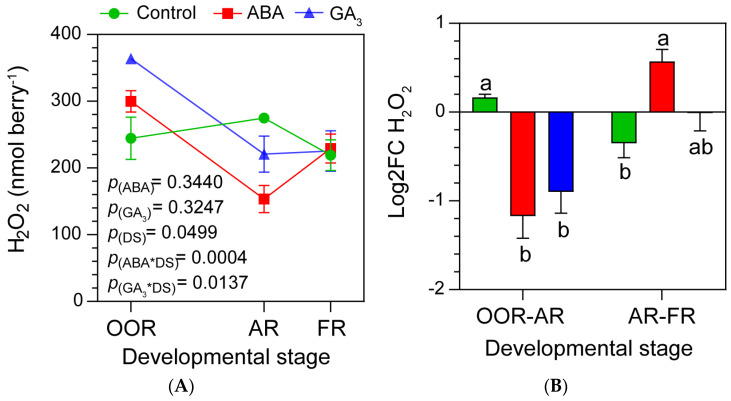
(**A**) Hydrogen peroxide (H_2_O_2_) content in whole berries according to the developmental stage and treatment. Values are means ± SEs, *n* = 3. Some errors cannot be shown because the SEs are smaller than the symbol. *p*(ABA), *p*(GA_3_), and *p*(DS): effects of ABA, GA_3_, and developmental stage, respectively; *p*(ABA*DS) and *p*(GA_3_*DS): interaction effects of factors. (**B**) Log_2_ fold change in H_2_O_2_ from OOR to AR (OOR-AR) and from AR to FR (AR-FR). One- and two-way ANOVA followed by Fisher’s LSD test were applied. Different letters indicate significant differences (*p* < 0.05).

**Figure 4 plants-13-02366-f004:**
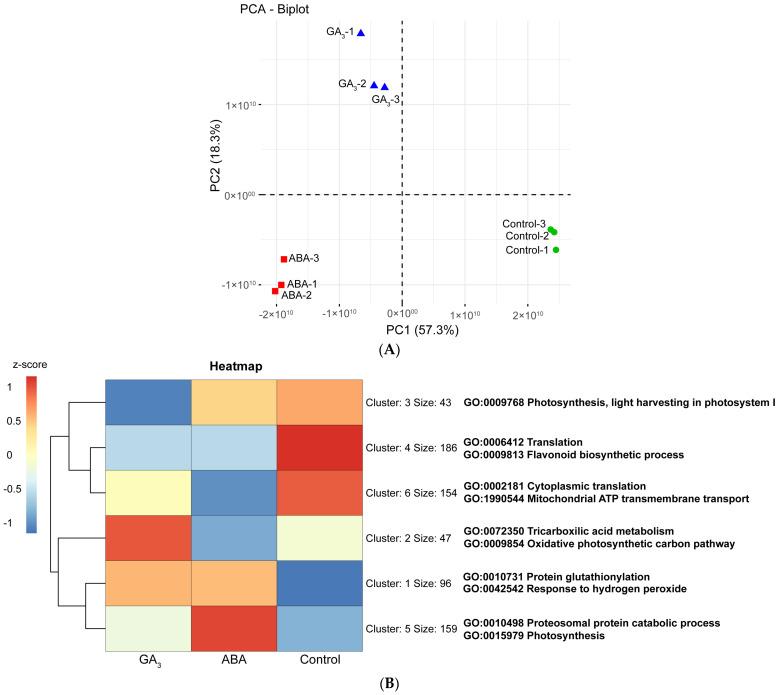
(**A**) Principal component analysis (PCA) of the 9 protein abundance datasets at the almost ripe stage (AR). Samples are clearly separated between control and treated samples (first principal component) and between ABA- and GA_3_-treated berries (second principal component). (**B**) k-means clustering heatmap of the 685 differentially abundant proteins (DAPs) in control, ABA-, and GA_3_-treated berry skins at AR. Size corresponds to the number of proteins grouped in each cluster. GO corresponds to the most representative Gene Ontology terms for biological processes of each cluster retrieved by STRING enrichment with a redundancy cutoff of 0. (**C**) PCA of variables measured at AR in control, ABA-, and GA_3_-treated berry skins. POX: Peroxidase domain-containing protein (E0CRP4); GPX-1: glutathione peroxidase-1 (F6HUD1); GPX-2: glutathione peroxidase-2 (F6H344); GPX-3: glutathione peroxidase-3 (A5AU08); APX-1: L-ascorbate peroxidase-1 (F6H0K6); APX-2: L-ascorbate peroxidase-2 (F6I106); SOD (Mn): superoxide dismutase (Mn) (F6HC76); SOD (Cu-Zn): superoxide dismutase (Cu-Zn) (D7SNA2); Gluta-PRX: glutaredoxin-dependent peroxiredoxin (D7TBK8); Thio-PRX: thioredoxin-dependent peroxiredoxin (D7TH54); PAL-1: phenylalanine ammonia lyase-1 (F6HNF5); PAL-2: phenylalanine ammonia lyase-2 (A5BPT8); C4H: trans-cinnamate 4-monooxygenase (A5BRL4); 4CL: 4-coumarate-CoA ligase (F6GXF5); 4CL-Like: 4-coumarate-CoA ligase-Like (F6GW98); CHI: chalcone-flavonone isomerase (F6HC36); F3′H: flavonoid 3′-monooxygenase (D7SI22); F3′5′H-2: flavonoid 3′;5′-hydroxylase-2 (F6HA82); ANS: anthocyanidin synthase (A2ICC9); UF3GT: UDP-glucose flavonoid 3-O-glucosyltransferase (D7SQ45); UF3;5GT: UDP-glucose: anthocyanidin 5;3-O-glucosyltransferase (A5BFH4); ANAT: anthocyanin acyltransferase (D7TU67). (**D**) Heatmap of the oxidative stress response, reactive oxygen species (ROS) production, and membrane degradation proteins in control, ABA-, and GA_3_-treated berry skins at AR. The values > 0 in the heatmap images indicate up-regulated proteins, while the values < 0 indicate down-regulated proteins.

**Figure 5 plants-13-02366-f005:**
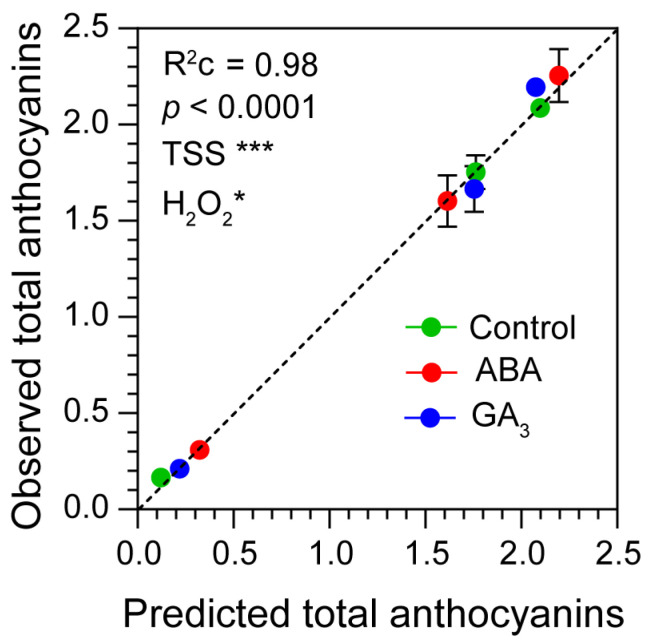
Observed total anthocyanins vs. predicted total anthocyanins. Linear mixed-effects model using total soluble solids (TSS, g berry^−1^) and hydrogen peroxide (H_2_O_2_, nmol berry^−1^) as fixed effects and treatment (control, ABA and GA_3_) as a random effect. R^2^c, conditional R^2^, represents the variance explained by the entire model. Significance codes: (***) *p* ˂ 0.001; (*) *p* ˂ 0.05.

## Data Availability

The raw data supporting the conclusions of this article will be made available by the authors on request.

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
