# Peer review of "Quantitative Proteomics Analysis of ABA- and GA3-Treated Malbec Berries Reveals Insights into H2O2 Scavenging and Anthocyanin Dynamics"

_plants, 2024, doi:10.3390/plants13172366_

Round 1

Reviewer 1 Report

Comments and Suggestions for Authors

Review Plants 3174781

ABA and GA3 applications are widely used in grapes to enhance berry color development and sugar accumulation. However, the mechanisms of their influence on the fruit ripening process remain largely unexplored Using physiological, biochemical and proteomic approaches, the authors thoroughly investigated the dynamics of H2O2 accumulation, changes in the berry skin proteome and phenolic compounds, as well as the total amount of soluble dry matter and the berry fresh weight during the ripening. Berries treated with ABA and GA3 exhibited reduced levels of H2O2 at almost ripe developmental stage (AR), which was attributed to the upregulation of ROS-scavenging proteins and the downregulation of ROS-generating proteins. Both treatments decreased anthocyanin content at this stage which correlated with a downregulation of the abundance of enzymes belonging to the anthocyanin biosynthesis pathway. Furthermore, ABA and GA3 treatments increased the protein abundance of the transcription factor Abscisic acid stress-ripening protein 2 (ASR2: F6GY46) which mediates glucose-ABA and glucose-GA crosstalk, modulating sugar accumulation and fruit ripening. The authors also revealed for the first time the capability of GA3 to induce ROS generation

These results are of significant interest for understanding the mechanism of action of hormones at the proteome and metabolome levels which is especially valued.  However, the text of the manuscript needs thorough editing as some of the authors' claims cause confusion.

Line 222 “a degradation of berry H2O2 either with ABA or GA3 applications from OOR to AR was found”.   ABA or GA3 do not degrade H2O2 directly

Lines 254-255 “ABA-treated berries mainly up-regulated the proteins sorted into proteasomal protein catabolic process…How can berries regulate proteins? Line 132

Lines 256-257 “As expected, this hormone also increased the abundance of proteins related to stress response suggested by the categories…”  and lines 263-265 “In addition, this hormone up-regulated the proteins related to the aromatic and L-serine and L-glycine amino acids metabolism suggested by the categories.” Please add “as” before “suggested” or rephrase these two passages.

The next two passages are also inconsistent:

Lines 340-343 “Contrary, cluster 2 grouped all the proteins up-regulated almost exclusively by the control (Figure 4D). In which, we found all the enzymes related to ROS production (NADH:ubiquinone reductase, LOX-1, LOX-2), membrane degradation (LOX-1, LOX-2, PLD-1, PLD-2 and PLC) and 5 GSTs (GST-1, GST-2, GST-7, GST-9 and GST-0).”

            Lines 346-348 ” Contrary to the expected due to the upregulation of enzymes involved in membrane peroxidation, control berries presented significantly less MDA content than the hormone treatments

Line 132 “Scheme of hormones applications” should be changed to “Scheme of hormone applications”

Line 384 “through the upregulation of structural and regulatory genes” It would be good to clarify which exact genes are meant.

I would also like to draw the authors' attention to the discussion of the results ( Lines .478-486). Without disputing the fact that “H2O2 may serve primarily as a signaling molecule rather than a toxic byproduct,” how can one explain the discrepancy between the reduced MDA levels and the increased NADH:ubiquinone reductase, LOX-1, and LOX-2 levels in control berry skins compared to hormone-treated samples, given the role of these enzymes in catalyzing membrane lipid peroxidation?

The significance of the results obtained for agricultural practice should also be explained.

Comments on the Quality of English Language

Editing is required

Reviewer 2 Report

Comments and Suggestions for Authors

The paper devoted to very interesting topic about effects of the exogenous ABA and GA treatments on grapevine fruit ripening.

Authors very clear shown significant effect of GA metabolite re-location during different stage of ripening.

The results are significant and interesting, while some contradiction because of authors use different part of the berry for extraction of protein, ROS and FW.  

More discussion are required on this contradictions.

Line 17: Endogenous ABA and GA are regulators of many developmental process and one of them is metabolite distribution during fruit maturation.

By the way, please, clarify do you mean exogenous ABA/GA?

 Line 57: “by promoting phloem area” ?? area can extend, not promoting.

Lines 74- 76: ROS production requires for all processes, not only for ripening.

Lines 310- 311: please, clarify what is Cu/Zn SOS and APX1 /APX2¿?

Plastids or cytoplasm¿?

Figure 4: why heatmap in B and D are in opposite orders?

Figure 4B: lines 303- 304: I am not sure these statements are 100% correct, based on fig 4 B.

Lines 315: “1 and two”?¿¿

Line 366: “Importantly, the observed differences among treatments were attributed to the effects of TSS and H2O2” ¿?? The effects of treatments were attributed to GA induced cell expansion, accompanied to ROS production on cell membrane and subsequent cell wall and membrane loosening (line 378- 379).

Line 589: how do you normalize H2O2 per berry if you use 150 mg of deseeds berry?

Line 606: only here is is written for the first time that proteome were isolated from berry skin, not from all berry, H2O2 were determined for the berry without seeds and some other parameters per whole berry. This contradiction should be mentioned in summery and in the results. 

Comments on the Quality of English Language

Minor corrections
